# Update on Suture Techniques in Corneal Transplantation: A Systematic Review

**DOI:** 10.3390/jcm11041078

**Published:** 2022-02-18

**Authors:** Luca Pagano, Haider Shah, Omar Al Ibrahim, Kunal A. Gadhvi, Giulia Coco, Jason W. Lee, Stephen B. Kaye, Hannah J. Levis, Kevin J. Hamill, Francesco Semeraro, Vito Romano

**Affiliations:** 1St. Paul’s Eye Unit, Royal Liverpool University Hospital, Liverpool L7 8XP, UK; haidershah@nhs.net (H.S.); kunal.gadhvi@doctors.org.uk (K.A.G.); giuliacoco@hotmail.it (G.C.); j.w.lee@liverpool.ac.uk (J.W.L.); s.b.kaye@liverpool.ac.uk (S.B.K.); 2Department of Biomedical Sciences, Humanitas University, Pieve Emanuele, 20090 Milan, Italy; 3Department of Medical and Surgical Specialties, Radiological Sciences, and Public Health, Ophthalmology Clinic, University of Brescia, 25121 Brescia, Italy; omar.alibrahim91@gmail.com (O.A.I.); francesco.semeraro@unibs.it (F.S.); vito.romano@gmail.com (V.R.); 4School of Medicine, University of Liverpool, Liverpool L69 3BX, UK; 5Department of Eye and Vision Science, University of Liverpool, Liverpool L69 3BX, UK; h.levis@liverpool.ac.uk (H.J.L.); kevin.hamill@liverpool.ac.uk (K.J.H.)

**Keywords:** graft, suture, PK, DALK, penetrating keratoplasty, lamellar keratoplasty, interrupted suture, continuous suture, running suture, nylon

## Abstract

Effective suturing remains key to achieving successful outcomes in corneal surgery, especially anterior lamellar keratoplasty and full thickness transplantation. Limitations in the technique may result in complications such as wound leak, infection, or high astigmatism post corneal graft. By using a systematic approach, this study reviews articles and conducts content analysis based on update 2020 PRISMA (Preferred Reporting Items for Systematic Reviews and Meta-Analyses criteria). The aim of this paper is to summarize the state of the art of corneal suturing techniques for every type of corneal transplant and patient age and also their outcomes regarding astigmatism and complications. Future developments for corneal transplantation will be also discussed. This is important because especially the young surgeon must have knowledge of the implications of every suture performed in order to achieve consistent and predictable post-operative outcomes and also be aware of all the possible complications.

## 1. Introduction

Suturing is the most vital skill for a corneal surgeon. The primary purpose of suture placement is to appose the wound edges in a normal anatomical position, aiding the natural healing process. The aim to minimize the formation of scar tissue, whilst simultaneously maximizing tensile strength, and preventing ingress of infectious agents into the globe. Effective suture placement also results in reduced postoperative astigmatism and long-term stability. A variety of techniques exist for suturing penetrating and lamellar keratoplasty; however, there is limited insight collating suturing methodology in different types of corneal transplants. This review summarizes the current knowledge of suture methodology in corneal transplantation and its effects on corneal transplant outcomes–providing a crucial tool of reference for the operating surgeon.

## 2. Corneal Wound Repair

Wound repair of the cornea is classically defined as a four steps process [1]. (Figure 1) Immediately after injury is first stage, known as the “latent” or “lag” phase. There are a series of cellular remodeling events leads to changes to tear film composition and expression of proteins required to initiate the wound repair process. This phase involves apoptosis of damaged cells, deconstruction of the adherens and gap cell-to-cell and the hemidesmosome cell-to-matrix adhesive devices, expression of extracellular matrix remodeling enzymes including matrix metalloproteinases, and expression of provisional matrix proteins such as fibronectin to fill the wound bed. In the second phase, “migration”, cells near the wound edge change from apical–basal polarity of front-rear polarity and migrate over the provisional matrix while depositing new extracellular matrix materials including laminins into the wound bed. The migratory process is driven by actin remodeling and force generation through temporary cell to matrix adhesive devices known as focal adhesions. Third, after migration is complete, the “proliferation” phase occurs, where corneal epithelial cells that have closed the wound divide and differentiate to restore the multi-layered structure that is the intact cornea. Finally, in the “maturation” phase, the cell-to-cell and cell-to-matrix adhesions reform and attach to anchoring fibrils. Restoration of these adhesive devices restores the corneal barrier function [2].

For deeper wounds, the process also involves activation of stromal keratocytes. Repair begins with apoptosis of keratocytes in the wound area, while other keratocytes transform into fibroblasts or myofibroblasts in response to expression or activation of transforming growth factor β (TGFβ) and platelet-derived growth factor (PDGF). Once activated, the fibroblasts synthesize collagen and fibronectin and reorganize the extracellular matrix tissue to provide a temporary replacement for the damaged stromal tissue. Myofibroblasts exert traction forces upon this provisional matrix, pulling the wound margins closer together. The provisional matrix laid down by these cell types is disorganized, and it is this excess matrix that leads to scarring and haze formation. The time during which the fibroblasts and myofibroblasts are active in the wound bed is the key determinant of the extent of post-wound scarring. Essential to the process of terminating the stromal aspect wound response is restoration of the epithelial basement membrane, a lamini- and collagen-rich structure laid down by corneal epithelial cells. Once the basement membrane is restored, expression of TGFβ and PDGF return to normal levels and activated cells undergo apoptosis. However, generation and maturation of the basement membrane can only occur once the epithelial layer is contiguous [3].

The length of time a wound takes to heal depends on the depth and width of the damaged area and the age and health of patient and will be slowed in the presence of infection. Suturing to bring wound margins closer together, which allows restoration of the epithelial basement membrane to occur more rapidly, can reduce the extent of scarring produced and more rapid resolution. Ideally the suture must appose the incised tissue edges in their normal anatomic position whilst providing adequate compression and minimal space for the scar tissue to bridge.

## 3. Surgical Tools

The choice of surgical instrument can substantially and substantively affect the efficiency and outcomes of a surgery. Although individual preference must be considered, surgeons must be aware of the advantages and limitations of the tools available to them. Prior to considering suture insertion, we must ensure the corneal tissue is handled in a controlled manner. Effective handling of tissue can be the difference between a successful corneal transplant and a post-operative complication. Techniques must be adjusted accordingly for every case, and each grasp must be carefully considered. Toothed forceps (ex Colibri forceps, Katena), for example, will grasp tissue well but can compromise tissue integrity. The alternate, notched forceps (ex Hoskin forceps, DTR medical) may be gentler on tissue; however, they may not grasp tissue as effectively. This is particularly problematic in oedematous tissue, where multiple grasps will macerate the wound edges and cause further swelling. In such cases, toothed forceps may be preferable.

Another important skill to master for a surgeon is dealing with essential hand tremor that can affect significatively suture construction and depth, given the small surgical space in ocular surgery.

To asses this issue, Hyung-Gon Shin et al. [4] in 2021 developed a new sensor-embedded automatic forceps, which detects when the device reaches the target suture depth and rapidly grasps the cornea to prevent error induced by human hand tremor. This tool is a hand-held device, so the learning curve is shorter compared to robot systems (ex Da Vinci surgical system).

## 4. Sutures Material

There are currently two main suture materials in use for corneal transplantation: nylon and Mersilene. Nylon sutures are easier to control, with minimal complications in the first two years after surgery. However, owing to its biodegradable nature, loosening and breakage, resulting in unpredictable astigmatism, can pose problems many years after surgery [5]. Mersilene is a polyester monofilament that is neither hydrolyzed nor degraded by ultraviolet light and is classed as non-biodegradable. Theoretically, its durability makes it more desirable; however, use is limited by its difficult handling, with a reported up to 5.5 times increased likelihood of handling complications compared with nylon sutures [6]. Mersilene use also carries a greater risk of complications within two years of surgery, including corneal infiltrates, metaplasia, and cheese-wiring [7].

Another important outcome to consider when choosing suture material is its effect on post-operative astigmatism and visual outcomes. This remains controversial. Although several studies suggest lower levels of astigmatism at the three-year point with the use of Mersilene sutures, no study has reported a statistically significant difference in BCVA at any follow-up period [5,7]. It has not been established whether the three-year findings are due to the superior abilities of Mersilene or the increased requirement for suture removal when using nylon sutures over this time period. It is also important to consider that, although Mersilene may result in a slight reduction in astigmatism, it requires more suture tension adjustments [5,7].

There is still debate about the role of Mersilene sutures as compared to nylon sutures in corneal transplant. Several studies conclude that Mersilene can be used effectively in corneal transplants [5,8,9], and though there is a possibility of reduced astigmatism, we must be wary of its difficulty handling, need for more frequent adjustments, and increased rate of complications within the first 2 years of surgery. Nylon sutures tend to be favored due to improved handling and lower initial complication rates, despite limitations from its biodegradable properties [7]. It is worth noting that evidence comparing these types of sutures have been on penetrating keratoplasty only, with limited evidence investigating their role in lamellar keratoplasty.

The role of silk sutures is limited in corneal transplantation. Although easy to handle and knot, certain properties limit its role in corneal transplantation. Two key components of the silk suture are fibroin and sericin. Sericin has been associated with Type 1 hypersensitivity reactions. Given the braided nature of silk, surface debris and micro-organisms may accumulate, resulting in inflammation and infection around the wound—a devastating complication for corneal grafts [10]. There is limited evidence for use of silk sutures in corneal transplantation specifically.

Non-absorbable sutures would be inappropriate for corneal transplantation as there would be a risk of localized inflammation and secondary delayed healing [10]. Early suture resorption would result in potential wound dehiscence. We could not identify studies that analyzed use of non-absorbable sutures alone; however, one study used a combination of 9-0 or 10-0 vicryl sutures in combination with 10-0 nylon sutures of non-absorbable sutures in 35 patients. Although adequate wound closure was maintained, with absorption within 6 weeks, use of the vicryl sutures was associated with delayed healing times [11].

The common role for absorbable sutures or silk sutures in corneal transplant surgery is for securing a Flieringa ring, which is beyond the scope of this review [12].

## 5. Suture Size

As with multiple facets of corneal suturing technique, selection of corneal suture size demonstrates the fine balance between ensuring adequate tension is placed on the wound, whilst also maintaining tissue integrity and minimizing astigmatism. Commonly, 10-0 nylon is the gauge of choice; however, alternatively, 11-0 may also be utilized [7,12]. When Mersilene sutures are used, 11-0 is used [12]. Previously, 9-0 vicryl sutures have been successfully used as compression sutures for the reduction of astigmatism following penetrating keratoplasty [13].

## 6. Suturing in Penetrating and Anterior Keratoplasty

The theory of suture placement dictates that the needle should enter the graft surface approximately 1.3 mm from the wound edge, reach 90–95% depth [14] through the donor tissue just anterior to the Descemet’s membrane, and finally exit in the recipient cornea at 1.3 mm, leaving equal distance from each side of the wound [15] (Figure 2). In this way, the inflammatory reaction produced is minimal, whilst holding the corneal wound in an appropriate anatomical position. Shallow sutures result in internal wound gape, whereas full depth corneal sutures may act as a conduit for surface micro-organisms to enter the eye. Unequal sutures result in wound override and consequent astigmatism [14].

There are different methods for tying the suture. A common technique includes a 3-1-1 closure, adjusting the tension before the second throw is made. Alternatively, the slipknot technique (1-1-1) allows for tension adjustment after the second throw, while the third throw ensures permanent tension. The first and second throws are in the same direction (during the second throw, it is therefore necessary to reach back around the knot to grasp the other end of the suture) so that the resulting knot is able to slide. Finally, the knot is retightened to the desired tension, and the final throw, in the opposite direction, locks the knot. Once all the sutures have been placed and the surgeon is satisfied with the tension, the ends can be trimmed with a microsurgical blade or Vannas scissors and knots buried. As yet, there is no clear evidence as to where it is better to bury the knot. Burying on the recipient side reduces the risk of wound dehiscence at the time of suture removal, whereas burying it at the donor side increases the distance between knot and the limbal vessels reducing inflammation and potentially rejection [16].

Cardinal suture placement is the starting point for any penetrating or anterior keratoplasty, and largely determines final corneal sphericity and, as a consequence, corneal astigmatism. It is therefore very important to analyze how to properly place them. The first cardinal suture, often considered the most difficult as the graft is unanchored, is typically placed at 12 o’clock. The second cardinal suture is typically placed at 6 o’clock, exactly 180 degrees from the first cardinal suture. Its positioning is an important determinant of final graft position, as equal bisection of the cornea can allow a more symmetrical placement of the other sutures, resulting in less astigmatism. Inadequate placement of this suture can have a knock-on effect, resulting in misalignment and asymmetry of final graft position. The third and fourth sutures are placed at the 3 and 9 o’clock positions, obtaining a diamond-shaped pattern. At this point, the architecture and configuration of the donor/recipient junction has been determined, irrespective of the placement of subsequent sutures.

It is also important to mention that cardinal sutures should be placed without waiting too long. In this moment, in fact, the eye is open, and any posterior pressure could potentially lead to dramatic consequences.

Once the cardinal sutures are in place, many different suture techniques have been proposed: interrupted sutures (SIS), combined interrupted and continuous sutures (CICS), single continuous running sutures (single CRS), and double continuous sutures (DCS) (Figure 3). It is important to be aware of the patterns of tension produced by different types of sutures, within a three-dimensional field. For example, simple interrupted sutures assume a circular shape, producing a tension that tends to cause inversion of wound edges. This is opposed to continuous sutures, which exert a more linear pattern of tension, and so invert wound edges in the case of a convex wound while tending to evert wound edges in the case of a concave wound. Symmetrical suture placement and tension result in more equal forces throughout the graft. Irregular sutures result from unequal depth, length, tension and non-radiality of suture placement.

### 6.1. Single Interrupted Suturing

The single IS technique represents the oldest of the techniques performed today. To obtain watertight closure, the minimum number of sutures required is thought to be eight; however, most surgeons perform 16 equidistant sutures, and some suggest a more cautious approach of up to 24 sutures. The increased number of sutures has the added benefit of a more circumferential distribution of tension, as opposed to the octagonal distribution obtained with the eight-suture option. This benefit, however, must be balanced against the risks associated with suture placement, as described in a later section. The main advantage of interrupted sutures is that it allows for quick suture replacement if, on post-op inspection, tension is not adequate. This suture generally employs 10-0 monofilament nylon sutures with a 160° single-curve 5.5 mm needle.

This is generally the first type of suture that a novice surgeon will adopt, since it is easier to perform, it does not carry the problems of breaking the suture mid-way, and it can easily be replaced in case the surgeon is not happy with the depth or orientation of the suture.

Astigmatic adjustment is performed by removing the suture at the steep meridian as guided by corneal topography. A single suture can be removed as early as six weeks postoperatively if it is felt to be too tight, contributing to high amounts of astigmatism, or if suture vascularization or infiltration occurs. Selective suture removal can continue until six months postoperatively to refine corneal shape. After six months, the wound is considered effectively closed, and all sutures can be removed. If the patient is happy with their vision, sutures could even be left in place indefinitely but should be removed in case of loosening, breaking, inflammation, infection, or vascularization.

### 6.2. Combined Interrupted and Single Continuous Suturing

A second method utilizes the combination of interrupted sutures and a CICS [17]. Generally, 12 interrupted sutures are followed by 12-bite continuous running sutures (CRS), although eight interrupted sutures and 16-bite continuous running sutures are also commonly employed.

The CS is accomplished by a 10-0 nylon suture in a clockwise fashion, superficial to the deeper interrupted sutures, with the first bite midway between 12 and 1 o’clock. It can be performed with radial bites equidistant between each of the interrupted sutures or by using an antitorque technique in which the apex of each bite in the donor cornea forms an isosceles triangle. The idea behind the antitorque technique is that it reduces the torsional effect that can be associated with radial bites [18]. Once all the bites are completed, the suture is tightened, removing the additional slack, and buried. Some surgeons perform intraoperative keratoscopy after the interrupted sutures are placed to adjust or replace them according to corneal astigmatism, and they generally repeat it after finishing the CS. Indeed, the main advantage of a CS is that tension can be redistributed evenly after tying the knot.

Postoperatively, interrupted sutures can be selectively removed as early as four weeks to reduce corneal irregularities, while the CS is generally removed 12–18 months following surgery.

### 6.3. Single Continuous Running Suture Technique

The single continuous running suture (CRS) is performed with a 24-bite of 10-0 nylon [19]. Similar to what we have described previously, after the four cardinal sutures, the surgeon starts the CRS between 12 and 1 o’clock, and the suture is run clockwise with 24 radial, 95% depth, evenly spaced bites. Once it is tightened, the cardinals are removed. The advantages are a faster surgical time and potential for suture adjustment intraoperatively and postoperatively. The disadvantages include risks of needle dullness or impaired wound integrity with only one improper bite, and the necessity to restart from the beginning if the suture breaks intraoperatively. The ideal time for postoperative adjustment is 3–6 weeks, as more complete wound healing increases the risk of suture breaks.

This is generally performed in theatre room for the risks of breaking the suture and consists in redistributing the tension from the steepest meridian to the flattest meridian.

### 6.4. Double Continuous Suturing

The DCS technique was first described in 1976 by Hofmann et al. [20] providing the benefits of a SCS, with the added security of a second CS. The technique entailed two 8-bite sutures at 90–95% depth. Further refinement of the technique [21] changed the depth of the second continuous suture. In fact, after the cardinal sutures, a 12-bite, 95% depth, clockwise running 10-0 nylon CRS is placed. A second clockwise suture (10-0 or 11-0 nylon or 11-0 Mersiline) is placed between each of the previous bites at 50% of the corneal depth to approximate the most superficial layers on both sides of the wound. It is passed at 50% depth because it minimizes the risk of breaking the deeper continuous suture and in addition it ensures a more uniform tension distribution at all cornea’s depths. It is usually recommended to remove the cardinal sutures only after both continuous sutures have been tightened, to avoid torsional movement of the graft. This technique requires more expertise than the other approaches; indeed, in addition to the previously described technical difficulties of a continuous suture, in DCS, it is also possible to break the first suture during a pass of the second one. However, the advantages include excellent long-term stability, rapid visual recovery and low levels of final astigmatism [22]. The superficial 10-0 nylon CRS is typically removed first at around 3 months, followed by removal of the more anterior suture at 12–24 months.

## 7. Effects of Suture Type on Astigmatism

The described suture techniques have been extensively explored for almost 40 years; however, the debate on which suture type results in less postoperative astigmatism has no clear answer yet. Suturing is a refined skill; in fact, corneal suturing is highly dependent on the manual skills of the surgeon, which inevitably will reflect their level of confidence and comfort with each specific technique.

In general, most studies tend to agree that the IS technique results in slightly higher astigmatism than the single CRS or DCS techniques [23]. In the 1990s, two studies compared the single CRS with the CICS and found that the former resulted in less astigmatism, fewer postoperative suture adjustments, and an earlier refractive stability (seven months earlier) [24]. As a counterpoint, single CRS was associated with a 7.2% risk of spontaneous wound dehiscence following suture removal. Further characterizing the different kinds of SCS, a comparison of three different techniques (torque, anti-torque, and no torque) did not find significant differences in astigmatism at three months and at six months follow-up [25]. More recently, a prospective study compared IS, single CRS, and DCS and found that both single CRS and DCS had lower degrees of astigmatism compared with IS during the first year of follow-up and at 18 months follow-up; when all sutures were removed, the DCS (3.60 ± 1.58 D) showed less astigmatism than the single CRS (5.65 ± 1.61 D) [26]. Many studies have tried to highlight differences between the single running and double running sutures. An adjustable single CRS was found to have better results at four months (1.7 ± 0.7 D) than a standard double running suture with no postoperative adjustment (5.4 ± 2.4 D, *p* < 0.01) [27]. Similarly, Ramirez et al. [28] found lower astigmatism at one year follow up with sutures still in place associated with the single CRS with postoperative adjustments, compared to DCS. These results are in agreement with two other studies that found no difference in the final astigmatism between the single CRS and the DCS once the sutures were removed, but DCS was associated with a faster restoration of visual function due to early stability [22].

Regarding the importance of intraoperative suture adjustment based on keratoscopy, a randomized control trial showed a significantly lower astigmatism at 1 and 6 months follow up compared to patients who did not have intraoperative adjustment [29]. Furthermore, none of the patients with sutures intraoperatively adjusted required postoperative adjustment, compared to the 77% of the control group. However, after suture removal at 15 months postoperatively, although astigmatism was lower in the intraoperative adjustment group (1.75 ± 1.04 D versus 2.23 ± 1.72 D [29]), there was not a statistically significant difference between the two groups.

A new prospective about different suture techniques in keratoplasty could be given from mathematical simulations of the cornea and the surgical factors: using a FEM (finite element analysis) model, it is possible to analyze how the depth and length of suture affect corneal curvature and distortion [30]. This could be a useful tool for developing new suturing techniques and improving refractive outcomes.

A study on 54 patients analyzing outcomes in deep anterior lamellar keratoplasty (DALK), comparing IS, single CRS, and CICS, found no significant difference among the three groups after removing the sutures at six months, and concluded that, due to earlier suture removal, the type of suturing technique used is not as important as it may be in a PK [31].

## 8. Effect of Graft Size on Astigmatism

The influence of graft size compared to recipient bed was originally studied in a retrospective analysis of 180 eyes operated on by a single surgeon comparing same-sized grafts versus 0.5 mm oversized ones [32]. Oversized grafts had higher astigmatism (6.44 D vs. 4.17, *p* < 0.01). However, four years later, a randomized control trial denied this association, concluding that oversizing up to 0.5 mm had no worsening effect on corneal astigmatism [33], and this was supported by a more recent work [34]. In 2003, a large prospective study [35] on 480 eyes who underwent PK with DCS investigated the influence of graft diameter (7 vs. 7.5 vs. 8 mm) on astigmatism. Smaller diameter grafts were associated with a higher degree of topographic irregularity; however, no differences were found in terms of astigmatism among the three groups, both before and after suture removal. No correlation has been found between astigmatism and either the time before suture removal or the corneal biomechanical parameters.

## 9. Mushroom vs. DALK vs. PK

Although a variety of techniques exist, as described above, there is limited, consistent evidence recommending which suture technique is superior for a certain type of graft. Evidence comparing types of sutures has been explored in penetrating keratoplasty only but has been adopted even for lamellar keratoplasties, where the evidence is limited.

The technique of suturing a mushroom graft should not differ significantly from a DALK; however, there is limited literature describing technical practices in detail. The full thickness, central button does not require any sutures, whereas the ‘mushroom cap’ is, in effect, treated as a lamellar keratoplasty. Surgery using either a double running 10-0 nylon suture or 16 interrupted 10-0 nylon sutures has been described [36]. Although sutures may be passed with 90–95% depth, passing the sutures through the anterior lamella may be performed instead, with the intention of allowing the posterior lamella to adapt accordingly. One point of note is, continuous suturing may not always be favorable in a mushroom graft, given its larger diameter, due to the limitation of recipient corneal tissue in the periphery.

As highlighted previously, the majority of influence on astigmatism is determined by the geometry of the wound construction and effective suturing, regardless of the type of graft. With this in mind, it is vital that all surgeons have knowledge of how their suture technique will affect the graft position. It is also advisable that operating surgeons performs a technique they are most comfortable and competent to perform.

It is recommended to perform a tomography on every graft at the 1, 3, and 6 months visits, which will be helpful to evaluate the postoperative astigmatism and can be extremely useful in case of novice surgeon to understand if they always have difficulties with specific suturing angles.

## 10. Endothelial Keratoplasty

### DSAEK Techniques in Unicameral Eyes

Establishing donor-recipient corneal adherence and minimizing the risk of graft dislocation is crucial in performing a successful Descemet’s stripping automated endothelial keratoplasty (DSAEK).

DSAEK and DMEK do not require sutures in most cases, and since anterior corneal curvature is preserved, astigmatism is much lower.

However, there are specific situations in which some suture might become beneficial. Unicameral eyes present a surgical challenge due to lack of intra-ocular compartmentalization and risk of dislocation into the posterior segment, resulting in possible complications such as corneal decompensation and proliferative vitreoretinopathy [37]. Although no consensus has been agreed upon, several techniques have been described.

One technique, as described by Price et al. [38], involved the insertion of the graft 80% into the anterior chamber, with the trailing 20% in the scleral tunnel or corneal limbal incision. The forceps were removed, and the cornea that remained within the incision held the donor button in place. A 10-0 nylon temporary fixating suture was passed from the recipient’s limbus at 11 o’clock, through the peripheral edge of the folded donor button and out through the overlying recipient cornea. The end of the donor tissue that had been held by the incision was then gently pushed into the anterior chamber. Air was then injected into the anterior chamber. In two out of three cases where this technique was performed, the fixation suture was removed (Figure 4).

Eguchi et al. described a technique in which a double armed 10-0 prolene suture that was originally designed for intraocular lens ciliary sulcus fixation acts as a lifeline for the lenticule DSAEK and as a dress-pin for the lenticule after DSAEK in vitrectomized aphakic eyes [37]. A similar technique has been described by Patel et al.; however, both arms are passed through the lenticule prior to insertion, risking suture entanglement and endothelial injury [39].

A wheel-spoke technique, described by Tanaka et al., has been described in a case of mydriatic bullous keratopathy but may be used for patients with total iris defects [40]. This technique involves passing three equally spaced 9-0 Prolene sutures radially through the cornea, limbus-to-limbus, in a pattern resembling wheel spokes in the 3 and 9 o’clock, 6 and 12 o’clock, and 4 and 10 o’clock directions. These acted as a safety net for the graft as it was unfolded by gently stroking the corneal epithelial surface. Air was used to tamponade the graft to the recipient cornea, and the Prolene sutures were removed. Post-operatively, the patient remained supine for three hours.

## 11. Suture Technique and Patient Age

Suture techniques and their management over time change according to the patient’s age. Differences in scleral and corneal elasticity, in wound healing, in the level of postoperative and suture-related inflammation, in infectious risk, in rejection rate, and in patient cooperation are all responsible for these changes. Furthermore, management of postoperative astigmatism through suture modulation has a different role in adult and pediatric patients [41]. Corneal sutures in paediatric patients pose specific difficulties compared to adults. First of all, there is a reduced space available for suturing since all the ocular strucures are smaller. Secondly, there is a significant association with anterior segment anomalies, especially in patients younger than 2 years old, that make suturing technically more challenging [41,42,43]. In addition, low rigidity of the sclera and cornea, a thinner and more pliable cornea and a higher posterior pressure lead to difficultly in apposing tissue and, consequently, it is more difficult to achieve adequate wound closure [41,44].

The increased posterior pressure also poses a specific risk of iris prolapse, expulsion of ocular content, and suprachoroidal hemorrhage during open-sky procedures such as PK, and some authors recommend using preplaced bridging mattress sutures in order to achieve rapid placement of the donor corneal button [41]. At the same time, in these thin corneas, an oversize of the donor graft of 0.5–1 mm is necessary to reduce the risk of excessive tissue compression and suture cheese-wiring in the recipient bed that would compromise wound closure [45]. Faster wound healing in children leads to tissue contraction, suture loosening, suture erosion and exposure, mucus accumulation and consequent increased risk of infections, corneal neovascularization, and graft rejection [41,46,47]. In addition, itchiness and irritation lead children to eye-rubbing and increase the risk of iatrogenic injuries, wound dehiscence, and infections [41,48,49]. It is estimated that infections can affect up to 50% of pediatric PKs, and most cases are due to suture-related complications [50], and endophthalmitis after suture removal, though rare, is more frequent in the pediatric population [51].

One study investigated the most frequent causes of paediatric unplanned hospital readmission within 30 days of discharge at a referral eye centre in the Middle East. Three out of the five major causes were suture-related, including loose corneal sutures in 17.6% of cases, wound dehiscence or leak in 8%, and persistent epithelial defect in 7.5%, the first, fourth, and fifth commonest causes, respectively. Among children with loose corneal sutures, 33% had undergone corneal repair for trauma, 30% had undergone PK, and 12% had cataract surgery with sutured corneal wounds [52].

Postoperative suture management is very challenging, and children are often unable to report symptoms of suture-related complications thus requiring frequent examinations [43]. Poor cooperation entails the need of examination under general anesthesia; however, from the age of 7–8 years old, sutures can often be removed at the slit lamp when the child shows good cooperation and is comfortable with the surgeon and with the environment [53].

The high rate of suture-related complications and their more difficult management in children has an impact on the suturing technique chosen. Options for wound closure might include 10/0 nylon running sutures, single interrupted sutures, or a combination of both. Running sutures would have the advantage of quick wound closure and more rapid suture removal. However, single interrupted sutures guarantee greater apposition of tissue margins, especially if some sutures break, loosen, or require removing, and are therefore more appropriate in these cases [53].

Some surgeons advocate for early suture removal in children, within the first 2–4 weeks after surgery, considering the rapid wound healing and the need of a tight control of postoperative inflammation with less risk of suture-related complications, corneal neovascularization, and graft rejection. However, benefits of early suture removal must be balanced with the risk of wound dehiscence, increased in children due to eye-rubbing [41]. Usually, PK suture removal can be safely performed about 4–6 weeks postoperatively in children less than 12 months old, 6–8 weeks postoperatively in those aged 1–2 years, around 8–12 weeks after surgery at 2–3 years old, 12–16 weeks postoperatively in children between 4 and 6 years old, and approximately at 16–24 weeks post-surgery in those aged 7–9 [54]. Loose sutures need immediate removal, and quick removal should also be performed for areas of corneal neovascularization. Selective suture removal to decrease corneal astigmatism is not practical and not useful in children since sutures are removed within a short period of time anyway [46,53]. Lastly, in children less than 3 years old, self-sealing scleral and corneal wounds commonly used in cataract surgery fail to be water-tight and have to be sutured [48,55]. These sutures can be removed as early as two weeks after surgery. Due to the need of general anaesthesia for suture removal in this age group, Bar-Sela et al. have advocated the use of 10-0 vicryl sutures in these cases. Vicryl sutures absorb postoperatively making their removal unnecessary and in a published case series they were associated with no suture-related complications at 6 months after surgery [47]. DSAEK sutures in paediatric cases can be removed as early as 2 weeks after surgery as well, allowing for prompt visual rehabilitation and early treatment of amblyopia in children [44] while timing for suture removal in DALK is similar to PK according to age and depending on the graft–host junction strength [53,56].

## 12. Suture Removal in Adults

There is no consensus in literature regarding corneal suture removal timing in adults, and different approaches are used based on surgeon experience.

In corneal transplant suture removal is done to assess post surgery astigmatism but preserving wound tightness.

In a survey made in 2003 by Richard MH Lee et al. [57], 137 British corneal consultant were asked to answer about PK corneal suturing removing time: the majority (41%) removed all corneal sutures at 1–2 years, 24% removing sutures at 1 year, while 25% had no specific routine for the removal of graft sutures. When removing the knot, 47/89 (53%) responded that they pulled the knot through the host side, and 30/89 (34%) through the donor, with 12/89 (13%) replying “either”. Seven of eighty-nine (8%) regularly pulled the knot through the interface [57].

## 13. Suture and Topical Treatment

As described above, corneal sutures lead to eye irritation and inflammation and lead to an increased risk of infections [41]. All patients are started on topical antibiotics and topical steroids on the first postoperative day. Topical antibiotics are usually continued until the corneal epithelium has healed, for approximately 7–14 days according to the type of surgery [53]. Frequency of instillation and strength of topical steroid varies with the type of surgery performed. Each selective suture removal has to be followed by 4 days of topical antibiotics [58] while suture removal due to suture-related infections is followed by topical broad-spectrum antibiotic continued until culture results are available and a targeted topical treatment can be started; application of topical steroids can be temporally suspended in these cases. Suture-related immune infiltrates should be treated with either intensification of topical steroidal treatment or with a short course of oral steroid if the epithelium is not intact [59].

Pediatric patients are commonly prescribed with prophylactic topical antibiotic treatment until all sutures are removed due to the higher risk of suture-related complications and infections and the inability of young children to clearly report symptoms of infections at their onset. Furthermore, the increased inflammatory response in paediatric cases requires topical steroid treatment to be prescribed at higher doses and for longer duration compared to adults and also tapered slowly.

Children are often prescribed topical cycloplegic drops to decrease the level of inflammatory response as well [53].

## 14. Contact Lenses Wear

Contact lens fitting in corneal transplant should be subdivided in therapeutic lens fitting and refractive contact lens fitting.

The latter is usually done to manage the residual irregular astigmatism in order to achieve the best visual acuity. Usually, rigid contact lens are used and are customized in order to adapt to the cornea shape and profile.

Therapeutic lens wearing are useful in the early post-op period in order to assess corneal epithelial defects that may cause patient discomfort.

Soft contact lens may be useful in dealing early small post operative wound leaks that can be managed with soft contact lens without further intervention.

Protruding suture that cannot be removed promptly by the surgeon can be managed by wearing a soft contact lens to alleviate patient discomfort [60].

## 15. Suture Related Complications

Suture related complications can be divided into intra-operative and post-operative.

Intraoperative complications are heavily influenced by the surgeon’s surgical skills, so a novel surgeon or a surgeon in training should bear attention to the most common problems facing corneal transplant suture.

Poor handling technique can lead to donor endothelial damage. Misplaced suture position can lead to iris incarceration and high risk of suture-related abscess in the post operative course. Improper suture tension can create undesirable astigmatism or donor–recipient mismatch, which can lead to difficulty in creating a watertight wound once suturing is completed.

One of the complications is forward movement of the lens–iris diaphragm, disrupting suturing by iris prolapse and creating a potential for lens damage or expulsion. The most dreaded complication creating this forward shift is a suprachoroidal hemorrhage [61].

Post-operative suture-related complications are a common reason for graft failure and a requirement for additional visits the operating room. The complications may present in the form of poor vision, a drop in vision, an uncomfortable eye, and a painful eye. Grossly, they can be divided into five areas: epithelial erosion around the suture, suture-related sterile subepithelial infiltrates, suture-related infectious keratitis, loose sutures, and dehiscence due to a loose suture or after suture removal [62].

### 15.1. Suture-Related Epithelial Erosion

Epithelial erosion at the suture site is common at any point following a keratoplasty and often asymptomatic [62]. However, patients may complain of a foreign body sensation, pain, or change in vision. In a study of penetrating keratoplasty, 10.8% of patients presented with suture-related epithelial erosion; however, only 32% of these patients were symptomatic, and more than half of them were contact lens wearers [62]. Management may entail removal of suture if sufficient time has passed or topical lubricants and prophylactic to aid healing and prevent infection if early in the post-operative period.

### 15.2. Subepithelial Immune Infiltrates at Suture Site

Subepithelial suture-related immune infiltrates are common at the site of suture entry into the corneal stroma [63]. Often presenting early in the post-operative period, these non infectious infiltrates are believed to arise from reaction to the suture material or contaminants along the suture such as talc from surgical gloves [64]. In a study of 361 grafts, there was a 9.4% incidence of non-progressive suture-related subepithelial infiltrates [62]. Often asymptomatic, these sub epithelial infiltrates rarely culture pathogens or progress to ulceration. Management usually involves removal of the responsible suture if possible and a short term course of antibiotics and topical steroids [62].

### 15.3. Suture-Related Infectious Keratitis

Corneal infections complicate 2–5% of transplants and a third of these infections results from suture complications [62,65,66,67,68]. Suture-related infectious keratitis may be catastrophic, frequently associated with corneal graft failure and sometimes with endophthalmitis and loss of the eye [69,70,71]. Risk factors include loose sutures, contact lens wear, and suture erosion, which may lead to broken sutures [65,66,71,72]. Infective organisms may vary, although a trend towards Gram-positive bacterial keratitis has been demonstrated [65,67,69,70]. Suture-related infectious keratitis management involves removal of the suture, culture of both the cornea and suture, and commencement of broad spectrum antibiotics until more tailored treatment can be delivered based on culture results. Infections related to deep or full thickness suture tracks may also act as a wick, leading to exogenous endophthalmitis, particularly when the suture is removed [66,73]. This stresses the importance of avoiding full thickness suture passes, minimizing the suture length that passes through the cornea during removal of suture, and the use of a sterile technique and antiseptic [15].

### 15.4. Loose Sutures

Loosening of sutures may develop after keratoplasty for four main reasons: (1) a tight suture becomes loose due to deturgescence of the graft; (2) insufficient re-epithelization at the suture tract resulting in potential microbial infection and stromal loss; (3) bio-degradation of the suture material; (4) absence/loss of Bowman’s in keratoconus resulting in cheese wiring of the suture [62]. Loose sutures may induce irritation and localized surface keratopathy, draw vessels from the limbus, and can lead to dehiscence or become a source of infection [62,69,70,71]. Loose suture management includes removal of the suture and consideration of re-suturing if at risk of dehiscence.

### 15.5. Suture Removal Related Dehiscence

Wound dehiscence after suture removal is a risk more common earlier in the post-operative healing process when wound healing is less complete; however, it remains a risk at any point after grafting. This is often a dilemma, as many of the aforementioned suture-related problems are resolved by suture removal. Christo et al. reported dehiscence in 8.3% of PKs and those requiring re-suturing in 2.4% following suture removal at a mean time of 10 months [62]. Management of a graft dehiscence involves re-suturing. To avoid this complication, interrupted sutures or double continuous sutures may be favored over single continuous ones. In single continuous sutures, breakage and removal of the entire suture length is more likely to induce dehiscence [15]. In addition, allowing more time between keratoplasty, especially in non-vascularized recipients, may allow more interface healing time [62]. Factors that may help avoid dehiscence of the wound include avoiding premature removal of sutures and avoiding the knot passing through the graft host junction at the time of removal to reduce the risk of disrupting the interface [15].

## 16. The Future of Wound Closure

As corneal grafting continues to evolve, so does the role of suturing. Corneal suturing may benefit from robotic input in the future. In 2009, in vitro experiments using the de Vinci surgical system (Si model) under remote teleoperated surgeon control demonstrated the ability to successfully place four cardinals followed by one 10/0 continuous running sutures [74].

However, several technical limitations were considered to be major hurdles to further investigations: poor visualization of the operative field, limited manoeuvrability of the instruments, and the absence of microsurgical instruments were considered by the authors to be hurdles to further investigations with the Da Vinci system.

Considering the recent technological improvements in the Da Vinci Surgical System, a new study conducted by Chammas et al. in 2017 demonstrated the feasibility of robot-assisted PK with the new Xi Da Vinci Surgical System.

Robot-assisted PK was successfully performed on 12 corneas, and the Da Vinci Xi Surgical System provided the necessary dexterity to perform the different steps of surgery [75].

Customized keratoplasty profiles is another field of growing interest within corneal grafting (Figure 5). With the advent of femtosecond assisted keratoplasty, customized graft profiles such as the mushroom keratoplasty have become more accessible. These profiles increase the surface area of apposition between graft and donor, potentially enhancing the wound stability and enabling earlier suture removal compared to grafts created with manual trephines and reducing dehiscence rate. Alio et al. demonstrated in a study of interface healing that there was significantly more scarring compared to manual trephination, which may infer a strong interface [76]. Various other custom femtosecond-assisted profiles have also been proposed, including zig-zag and decagonal profiles [77,78,79,80]. This remains a developing field of corneal surgery, which may enable earlier suture removal and reduce late suture-related complications.

Sutureless corneal grafting surgery, although not suitable for full thickness keratoplasty, has been demonstrated in superficial lamellar keratoplasty to treat conditions such as anterior stromal haze [81,82]. Surgery may be performed with a microkeratome or a femtosecond laser to complete the desired donor and host cut [82,83]. Although overlay sutures may temporarily be required, most cases can be managed post-operatively with a bandage contact lens until healing is complete avoiding the need for post-operative suture management.

Ongoing materials research has resulted in new suture materials and manufacturing processes such as melt spinning of a block copolymer to create a monofilament fibre that is comparable in strength to monofilament suture materials in current clinical use, with the advantage that they are less costly to produce [84]. Other new bioabsorbable suture materials include self-reinforced poly-l-lactide (SR-PLLA), which has been found to have longer retention of tensile strength when compared with polyglyconate and polydioxanone in vitro [85], and lactide-epsilon-caprolactone copolymer (P[LA/CL]), whose degradation is not affected by changes in pH [86]. Recent advances in suture technology include coating of polyglactin sutures with both bioactive glass and anti-bacterials. Polyglactin sutures with bioactive glass coating have been shown to develop bone-like hydroxyapatite crystal formation around the suture when immersed in simulated body fluid [87]. The hydroxyapatite layer can become part of a 3-D scaffold for further tissue engineering applications [87,88,89]. Silver impregnation of the bioactive glass coating can impart antibacterial properties to the suture as can coating of the suture with triclosan [90]. Recent investigations of silk fibres, which are far more inert than previously believed [91], have revealed that it, too, has potential for tissue engineering by addition of growth or adhesion factors to silk’s multitude of different side chains [92].

## 17. Glue Sealant

Tissue adhesives provide promising substitutes for sutures in ophthalmic surgery. Ocular adhesives are not only intended to address the shortcomings of sutures but also designed to be easy to use, and can potentially minimize post-operative complications.

Grissel Trujillo-de Santiago et al. [93] describe the types of corneal sealant currently available and classified them based on their chemical structure: synthetic adhesives based on cyanoacrylates, linear polyethylene glycol (PEG) derivatives, and dendrimer-based adhesives.

Biological based sealant are classified in protein-based adhesives: fibrin glue, serum albumin glue, and collagen and gelatin-based glue, as well as polysaccharides such as chondroitin sulfate, dextran, and hyaluronic acid.

These materials are made up by polymers that are applied as fluids at the ocular wound site and are chemically or physically activated to bind and hold tissues.

Ocular adhesives not only prevent the patient and the surgeon from experiencing the drawbacks of sutures but also can potentially offer important functionalities such as the feasibility to match the biomechanical properties of the native tissue, so the wound healing progresses without limiting tissue movement or affecting its function [94].

Cyanoacrylate and fibrin glue are the two most common tissue adhesives in ophthalmology: the first one shows great strength; however, it is not recommended in corneal transplantation suture because of its toxic nature, lack of transparency, and flexibility.

Fibrin glue shows good transparency and biocompatibility but has low adhesion strength.

Moreover, viral transmission is a serious concern in fibrin-based sealant even though viral screening is always performed.

The ideal glue sealant should be being nontoxic and biocompatible in patients, an ideal adhesive that can replace sutures in corneal transplantation requires to being optically transparent to permit vision and having adequate adhesive strength to hold corneal grafts in place.

Xuan Zhao et al. [94] in a study developed a new type of synthetic polymer bioadhesive hydrogels made by dexthrane and showed its efficacy in vitro and lamellar transplant in an animal model (rabbit).

This new kind of sealant showed good biocompatibility in vitro and also good strength, enduring a pressure of 280 mmHg.

In vivo animal model the cornea preserved a smooth surface and complete curve with no visible scar and epithelial ingrowth on the donor–host interface.

The donor graft and recipient cornea remained transparent without excessive inflammation, and epithelial defects present ad day one after surgery were completely healed after 7 days [94].

## 18. Method of Literature Research and Selection Criteria

In December 2020, a systematic literature review was started using PubMed and its subsidiary MEDLINE, EMBASE, and Web of Science for the components of this review. The studies examined were identified on PubMED using the following keywords ‘Corneal transplant’, ‘penetrating’, ‘keratoplasty’, ‘lamellar’, ‘DSAEK, ‘PK’, ‘DALK’, ’mushroom keratoplasty’ ‘opthtalmic instrument’, ‘corneal suture’, ‘corneal wound AND wound dehiscence’, ‘(post surgery astigmatism) AND (astigmatism and suture)’, ‘corneal suture complication’, ‘continuous’, ‘interrupted’, ’running’, ‘nylon cornea’, ‘mersilene cornea’, ‘eye robotic surgery’, and ‘Suture-related complications keratoplasty’, and the publications extracted were dated between January 1960 and November 2021. The last research query was on 21 November 2021.

Several textbooks were also reviewed and referenced accordingly.

From the search queries, we recovered a total of over 20,000 papers. Given the volume of literature recovered, our inclusion criteria included publications in the English language. We included controlled clinical trials (CCT), prospective randomized studies, comparative studies, and reviews. We chose only articles available in full text; with this criteria, 5589 were recovered.

An AML (adaptive machine learning software) (ASreview https://github.com/asreview accessed on 21 November 2021) was used to screen the papers retrieved. (See flow diagram below Figure 6).

The authors chose the articles based on their relevance and significance for each topic: a total of 88 papers were selected.

## Figures and Tables

**Figure 1 jcm-11-01078-f001:**
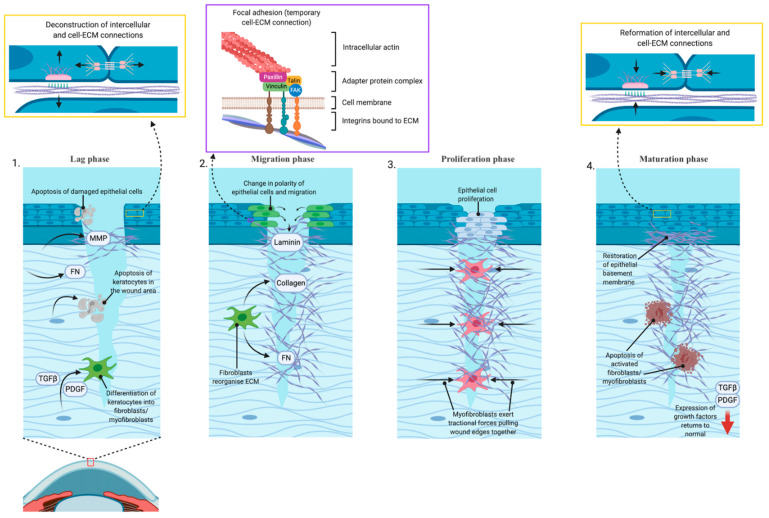
The corneal wound healing process summarized as four steps. (1) Lag phase immediately after injury: cellular remodeling and expression of matrix metalloproteinases (MMP) and fibronectin (FN). Deeper stromal keratocytes undergo fibroblastic change via transforming growth factor β (TGFβ) and platelet-derived growth factor (PDGF). (2) Migration phase. Deposition of extracellular matrix (ECM) materials and reorganization of the provisional matrix. Temporary connections form via cell–ECM focal adhesions. (3) Proliferation phase. Epithelial cell proliferation with continued ECM secretion and traction to close wound. (4) Maturation phase. Restoration of epithelial basement membrane, decreased cellular activity, and apoptosis.

**Figure 2 jcm-11-01078-f002:**
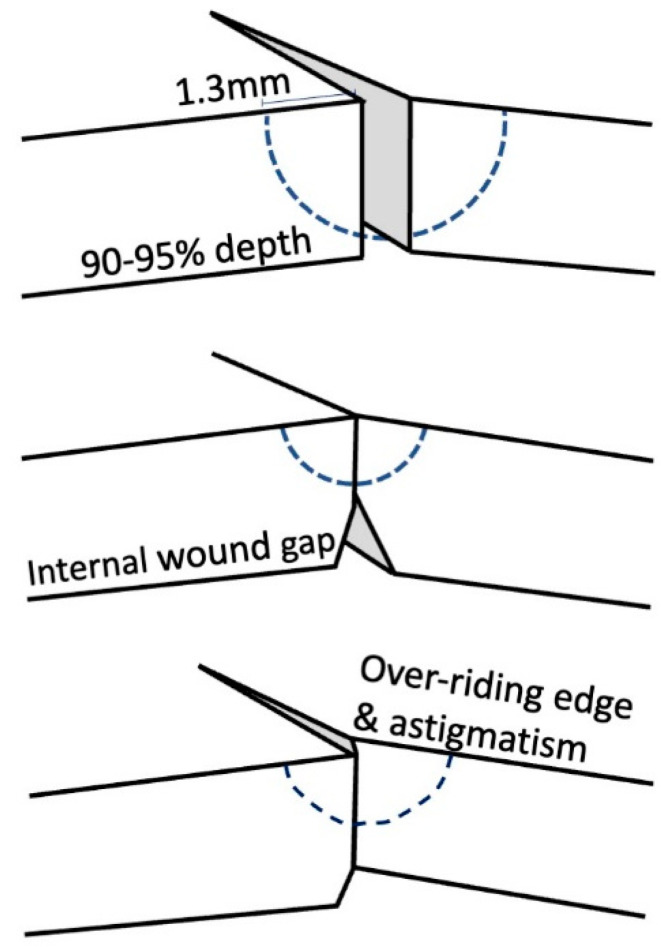
Demonstration of suture depth. 1. Ideal suture depth. 2. Shallow suture. 3. Unequal sutures.

**Figure 3 jcm-11-01078-f003:**
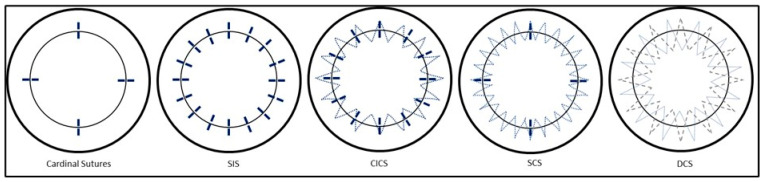
Types of corneal sutures. SIS = single interrupted sutures, CICS = combined interrupted and continuous sutures, Single CRS = single continuous running suture, DCS = double continuous suture.

**Figure 4 jcm-11-01078-f004:**
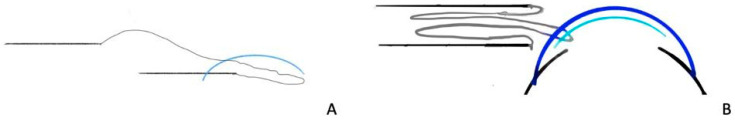
Unicameral Dsaek delivery with suturing of the posterior lamellar graft. (**A**) A double ended 10/0 straight prolene needle (STC-6) is passed through the donor (**B**) both ends of the suture are pre placed through the cornea opposite to the main incision; then, only after the delivery and centration of the graft, the suture is tightened.

**Figure 5 jcm-11-01078-f005:**
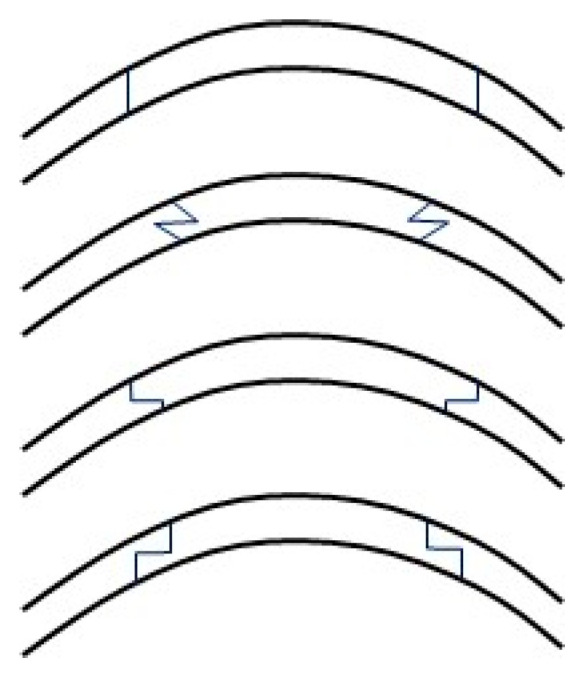
Variations in wound shapes for penetrating keratoplasty. From top to bottom; conventional, zigzag, mushroom, and top-hat morphology.

**Figure 6 jcm-11-01078-f006:**
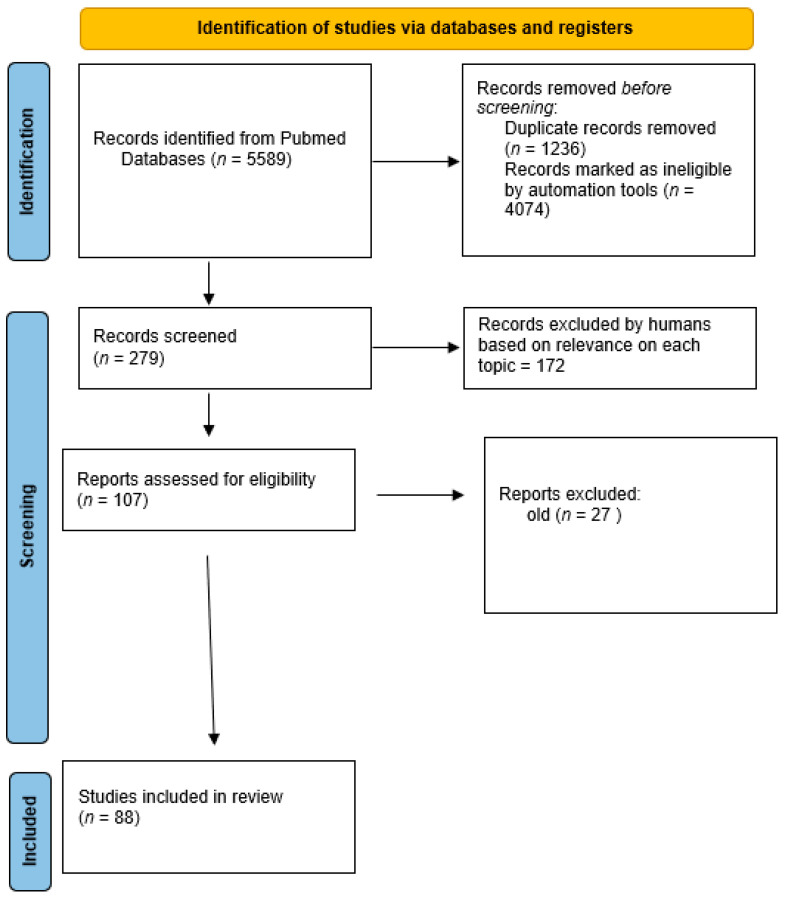
Flow diagram regarding paper selection process using Prisma 2020 statement [95].

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
