# Peer review of "Update on Suture Techniques in Corneal Transplantation: A Systematic Review"

_jcm, 2022, doi:10.3390/jcm11041078_

Round 1

Reviewer 1 Report

This review is a comprehensive guide on the use of sutures in corneal surgery, and should be a valuable reference for new surgeons aiming to understand the importance of suture type, placement, adjustments, complications, etc. It is a well-written and appropriate review of the literature on this topic, providing valuable insights. I have highlighted below a number of areas which the authors should consider in order to improve this review.

Figure 1 is a very nice overview, however the names of the phases in the text do not always match the names in the figure caption. Please make these consistent. Also, could the names of the phases be included in the figure, above or below each of the 4 blocks?

Line 95-102: no reference for these tools, can the authors at least give a product name and supplier? Are photos available?

Line 108: what are the robot systems, please explain and give appropriate reference (perhaps just reference here, as it is described later in the review)

Regarding suture materials, what about silk sutures and absorbable sutures? Have these been used in corneal surgery and what is their role?

What about suture size/gauge? What are the most common gauges used and why? Some references can be given.

Font size of text in Figure 2 should be increased for readability.

Lines 150-153: it is hard to visualize what is meant in this description. Can a more detailed description be given, including terminology definitions? A simple diagram of the steps would be helpful, or if this has been given elsewhere please provide the reference.

Should the first cardinal sutures be placed within a given time limit? What is usual for a beginner versus an experienced surgeon?

Line 173: IS should be ’SIS’ to be consistent with figure 3

Line 177: what do you mean by ‘inversion of wound edges’ ?

This review is, according to the abstract, aimed at helping and guiding new surgeons. The descriptions of the various suturing techniques, which is an important part of this review, however, does not give a guide to the reader about which technique to use in which situations and indications? It is mentioned that each surgeon chooses the method they are most comfortable with, but in the beginning a new surgeon is not comfortable with any method. Based on the authors experience and the literature, I would suggest to include in this section on different suturing techniques a better guidance for new surgeons on how to choose the method to use based on particular clinical context.

Line 228: how can a continuous suture be adjusted postoperatively? It seems that interrupted sutures can easily be adjusted but I am unclear about continuous.

Line 235: why does the first suture in DCS go to 80% depth while the CRS goes to 95% depth?

Line 238: why is the second DCS suture at 50% depth, and why would Bowman’s layer be involved, which is much more superficial than 50%, it would be around 10% depth.

Line 243: ¨The deep…..is typically removed first¨, when specifically?

Line 272: ’randomised control’ should be ‘randomized controlled trial’

Line 272-274: What type of ¨intraoperative suture adjustment¨? Do you mean keratometry/keratoscopy? 

Line 285: ’he’ should be ‘The’ at the start of the sentence.

Paragraph line 285-296: Include a separate title like ¨Effect of graft size on astigmatism¨

Lines 313-317: how should a new surgeon evaluate their suturing technique? When should they measure topography and astigmatism in the patient?

Is there any evidence on whether femto laser gives better wound apposition and therefore lower astigmatism, regardless of suture technique, compared to manual surgery?

Section on endothelial keratoplasty: It should be stated that DSAEK and DMEK do not require sutures in most cases, and anterior corneal curvature is preserved thus astigmatism is much lower.

Perhaps a drawing to illustrate the procedure for unicameral eyes would be helpful for the reader.

Line 360: the cornea in children and infants should have the same thickness or even be slightly thicker than in adults, so the statement is confusing.

I like the section on considerations and choice of sutures, removal, etc. for children. It seems that patient characteristics (even adults) can influence the type and timing of sutures and suture removal. It also may be the case that in lower income countries, a different set of criteria are guiding the choice of sutures and suture management. It would be very interesting to consult corneal surgeons in those countries to hear their perspectives.

Line 440: astigmatism is the most common postoperative complication so should that entire section go under complications?

Because this is a review to aid new surgeons, I would suggest that intraoperative complications secondary to sutures are also included, not only postoperative complications.

Can the authors describe the considerations of contact lenses in combination with sutures, when can they be worn after surgery, special considerations, etc.

Is there any role for adhesives/glue in combination with sutures? Are there examples of this in the literature?

Author Response

Reviewer 1

This review is a comprehensive guide on the use of sutures in corneal surgery, and should be a valuable reference for new surgeons aiming to understand the importance of suture type, placement, adjustments, complications, etc. It is a well-written and appropriate review of the literature on this topic, providing valuable insights. I have highlighted below a number of areas which the authors should consider in order to improve this review.

Figure 1 is a very nice overview, however the names of the phases in the text do not always match the names in the figure caption. Please make these consistent. Also, could the names of the phases be included in the figure, above or below each of the 4 blocks?

The figure caption has been updated and now each phase matches with the text. Figure has been updated as well to include the 4 different phases.

Line 95-102: no reference for these tools, can the authors at least give a product name and supplier? Are photos available?

Product name and supplier were added.

Line 108: what are the robot systems, please explain and give appropriate reference (perhaps just reference here, as it is described later in the review)

A reference was added

Regarding suture materials, what about silk sutures and absorbable sutures? Have these been used in corneal surgery and what is their role?

What about suture size/gauge? What are the most common gauges used and why? Some references can be given.

We added a paragraph regarding sik and absorbable sutures

Font size of text in Figure 2 should be increased for readability.

Updated

Lines 150-153: it is hard to visualize what is meant in this description. Can a more detailed description be given, including terminology definitions? A simple diagram of the steps would be helpful, or if this has been given elsewhere please provide the reference.

A better description of how the knot is tied has been given in the text.

Should the first cardinal sutures be placed within a given time limit? What is usual for a beginner versus an experienced surgeon?

Added a parargraph about posterior pressure and open eye.

Line 173: IS should be ’SIS’ to be consistent with figure 3

Modified

Line 177: what do you mean by ‘inversion of wound edges’ ?

We mean that considering how tension is distributed in case of a single suture the wound edges will tend to go inward.

We modified the text to make it more clear.

This review is, according to the abstract, aimed at helping and guiding new surgeons. The descriptions of the various suturing techniques, which is an important part of this review, however, does not give a guide to the reader about which technique to use in which situations and indications? It is mentioned that each surgeon chooses the method they are most comfortable with, but in the beginning a new surgeon is not comfortable with any method. Based on the authors experience and the literature, I would suggest to include in this section on different suturing techniques a better guidance for new surgeons on how to choose the method to use based on particular clinical context.

Regarding single interrupted suture we added a paragraph to recommend this technique to novice surgeons. “This is generally the first type of suture that a novice surgeon will adopt, since it is easier to perform, it doesn’t carry the problems of breaking the suture mid-way and can easily be replaced in case the surgeon is not happy with the depth or orientation of the suture.”

Line 228: how can a continuous suture be adjusted postoperatively? It seems that interrupted sutures can easily be adjusted but I am unclear about continuous.

This is generally performed in theatre room for the risks of breaking the suture and consists in redistributing the tension from the steepest meridian to the flattest meridian.

Line 235: why does the first suture in DCS go to 80% depth while the CRS goes to 95% depth?

That was a mistake, we corrected and changed it to 95% depth. There is no reasoning for it to be different than a CRS.

Line 238: why is the second DCS suture at 50% depth, and why would Bowman’s layer be involved, which is much more superficial than 50%, it would be around 10% depth.

We added “It is passed at 50% depth because it minimizes the risk of breaking the deeper continuous suture and in addition it ensures a more uniform tension distribution at all cornea’s depths”.

We modified the bowman layer and changed it with more superficial layers.

Line 243: ¨The deep…..is typically removed first¨, when specifically?

There was a mistake, the first suture to be removed is the superificial wich is generally removed at 3 months.

We changed the text accordingly. The superficial 10-0 nylon CRS is typically removed first at around 3 months, followed by removal of the more anterior suture at 12–24 months.

Line 272: ’randomised control’ should be ‘randomized controlled trial’

Corrected.

Line 272-274: What type of ¨intraoperative suture adjustment¨? Do you mean keratometry/keratoscopy? 

Exactly, we modified the text to make it clearer

Regarding the importance of intraoperative suture adjustment based on keratoscopy, a randomised control trial showed a significantly lower astigmatism at 1 and 6 months follow up compared to patients who didn’t have intraoperative adjustment

Line 285: ’he’ should be ‘The’ at the start of the sentence.

Corrected

Paragraph line 285-296: Include a separate title like ¨Effect of graft size on astigmatism¨

Added

Lines 313-317: how should a new surgeon evaluate their suturing technique? When should they measure topography and astigmatism in the patient?

Added a sentence on this topic: It is recommend to perform a tomography on every graft at the 1, 3 and 6 months visits which will be helpful to evaluate the postoperative astigmatism and can be extremely useful in case of novice surgeon to understand if they always have difficulties with specific suturing angles.

Is there any evidence on whether femto laser gives better wound apposition and therefore lower astigmatism, regardless of suture technique, compared to manual surgery?

We found no evidence regarding that.

Section on endothelial keratoplasty: It should be stated that DSAEK and DMEK do not require sutures in most cases, and anterior corneal curvature is preserved thus astigmatism is much lower.

Thanks, we added this comment.

Perhaps a drawing to illustrate the procedure for unicameral eyes would be helpful for the reader.

We added a new figure for unicameral eye

Line 360: the cornea in children and infants should have the same thickness or even be slightly thicker than in adults, so the statement is confusing.

We rephrased the sentence, when we are talking about smaller ocular structures we are not referring to cornea thickness but rather to the corneal diameter and the space in the anterior chamber.

I like the section on considerations and choice of sutures, removal, etc. for children. It seems that patient characteristics (even adults) can influence the type and timing of sutures and suture removal. It also may be the case that in lower income countries, a different set of criteria are guiding the choice of sutures and suture management. It would be very interesting to consult corneal surgeons in those countries to hear their perspectives.

We agree that these recommendations are significantly influenced by the settins each surgeon is facing. It is definitely an interesting point of reflection.

Line 440: astigmatism is the most common postoperative complication so should that entire section go under complications?

We agree that astigmatism could be considered the most common postoperative complication, but since it is “expected” in a way, we think that the postoperative management of the astigmatism is actually part of the procedure itself. That’s why we preferred to dedicate its own section to it.

Because this is a review to aid new surgeons, I would suggest that intraoperative complications secondary to sutures are also included, not only postoperative complications.

We added a paragraph for intraoperative complications

Can the authors describe the considerations of contact lenses in combination with sutures, when can they be worn after surgery, special considerations, etc.

We added a paragraph for contact lenses

Is there any role for adhesives/glue in combination with sutures? Are there examples of this in the literature?

We added a paragraph for adhesives/glue

Reviewer 2 Report

In their manuscript entitled “Update on suture techniques in corneal transplantation: a systematic review”, authors Pagano et al. provide an overview of important aspects to consider when suturing anterior lamellar or full-thickness corneal grafts. The article is geared towards providing evidence-based knowledge on this subject to (not only) novice surgeons. It may therefore indeed be a valuable resource for clinicians. To make this work publishable, however, the authors would need to adopt a more rigorous systematic approach to their literature, or at least more thoroughly describe their technique for literature search and analysis (see major point for improvement below).

Major point for improvement:

Line 21-22 (and title, and Methods): The authors state that they used “modified PRISMA criteria” to produce their “systematic” review. However, it is not described in the methods section how the PRISMA standard has been modified. The authors should include details of their checklist/items/scoring method, and provide a flow diagram of the selection process of papers that were included in the analysis.

Minor points for improvement:

Lines 54 and 79: Unlike with epidermis, epithelial cells of the cornea are not called keratinocytes. They are corneal epithelial cells.

Line 234: Another variant of the DCS technique was described one year earlier by Hoffmann (Hoffmann F. Suture technique for perforating keratoplasty. Klin Monbl Augenheilkd 1976;169:584-90).

Line 235: This technique (by Hoffmann; see comment above) entails two 8-bite sutures at 90-95 % depth.

Lines 236-237: It is usually recommended to remove the cardinal sutures only after both continuous sutures have been tightened, to avoid torsional movement of the graft.

Line 272: “…randomised controlled trial…”?

Lines 277-278: If there was no statistically significant difference between the groups, then it cannot be stated that astigmatism was lower in the intraoperative adjustment group.

Line 526: Should probably read “apposition” instead of “opposition”.

The authors may want to consider adding a small paragraph dedicated to the timing of suture removal in adults, as there seems to be not much consensus on this.

While the relevance of a good suture technique in achieving good graft placement and low astigmatism is undisputed, the authors may wish to add a sentence to remind us that the geometry of the wound bed is also of great importance.

Author Response

Reviewer 2

In their manuscript entitled “Update on suture techniques in corneal transplantation: a systematic review”, authors Pagano et al. provide an overview of important aspects to consider when suturing anterior lamellar or full-thickness corneal grafts. The article is geared towards providing evidence-based knowledge on this subject to (not only) novice surgeons. It may therefore indeed be a valuable resource for clinicians. To make this work publishable, however, the authors would need to adopt a more rigorous systematic approach to their literature, or at least more thoroughly describe their technique for literature search and analysis (see major point for improvement below).

Major point for improvement:

Line 21-22 (and title, and Methods): The authors state that they used “modified PRISMA criteria” to produce their “systematic” review. However, it is not described in the methods section how the PRISMA standard has been modified. The authors should include details of their checklist/items/scoring method, and provide a flow diagram of the selection process of papers that were included in the analysis.

We updated and clarified the criteria used and added a flow diagram of the selection process of papers that were included in the analysis

Minor points for improvement:

Lines 54 and 79: Unlike with epidermis, epithelial cells of the cornea are not called keratinocytes. They are corneal epithelial cells.

Thanks for the correction, we modified the text accordingly.

Line 234: Another variant of the DCS technique was described one year earlier by Hoffmann (Hoffmann F. Suture technique for perforating keratoplasty. Klin Monbl Augenheilkd 1976;169:584-90).

Thanks, we added this reference in the text

Line 235: This technique (by Hoffmann; see comment above) entails two 8-bite sutures at 90-95 % depth.

Lines 236-237: It is usually recommended to remove the cardinal sutures only after both continuous sutures have been tightened, to avoid torsional movement of the graft.

Thanks, we modified the text accordingly.

Line 272: “…randomised controlled trial…”?

Corrected

Lines 277-278: If there was no statistically significant difference between the groups, then it cannot be stated that astigmatism was lower in the intraoperative adjustment group.

We changed the sentence to stress that there was not a statistically significant difference.

Line 526: Should probably read “apposition” instead of “opposition”.

Modified

The authors may want to consider adding a small paragraph dedicated to the timing of suture removal in adults, as there seems to be not much consensus on this.

We added a pargrapha on timing of suture removal

While the relevance of a good suture technique in achieving good graft placement and low astigmatism is undisputed, the authors may wish to add a sentence to remind us that the geometry of the wound bed is also of great importance.

Added to the text: “As highlighted previously, the majority of influence on astigmatism is determined by the geometry of the wound construction and effective suturing, regardless of the type of graft.”